



# Weather pattern dynamics over Western Europe under climate change: Predictability, Information Entropy and Production

Stéphane Vannitsem[1]

[1]Royal Meteorological Institute of Belgium

**Correspondence:** Stéphane Vannitsem (Stephane.Vannitsem@meteo.be)

**Abstract.**

The impact of climate change on weather pattern dynamics over the North Atlantic is explored through the lens of the information theory of forced dissipative dynamical systems.

The predictability problem is first tackled by investigating the evolution of block-entropies on observational time series
of weather patterns produced by the Met Office, which reveals that predictability is increasing as a function of time in the observations during the 19th and beginning of the 20th Century, while the trend is reversed at the end of the 20th century and beginning of the 21st Century. This feature is also investigated in the 15-member ensemble of the UK Met Office CMIP5 model for the 20th and 21st centuries under two climate change scenarios, revealing a wide range of possible evolutions depending on the realization considered, with an overall decrease of predictability in the 21st century for both scenarios.

Lower bounds of the information entropy production is also extracted providing information on the degree of time-asymmetry and irreversibility of the dynamics. The analysis of the UK Met Office model runs suggests that the information entropy production will increase by the end of the 21st century, by a factor of 10 % in the RCP2.6 scenario and a factor of 30-40 % in the RCP8.5 one, as compared to the beginning of the 20th century. This allows for making the conjecture that the degree of irreversibility is increasing, and hence heat production and dissipation will also increase under climate change, corroborating
earlier findings based on the analysis of the thermodynamic entropy production.

## 1 Introduction

The climate system is a forced dissipative system, whose forcing depends on time. Among the components of this forcing, one can mention the obvious natural forcing (solar, volcanic eruptions...), but one of the most important forcing in the recent decades is the anthropogenic forcing known to strongly affect the climate system (e.g. Lovejoy , 2014; Hébert and Lovejoy ,
2018; Ghil and Lucarini , 2020; IPCC , 2021). This anthropogenic forcing is inducing a rapid global increase of temperature as amply illustrated in the IPCC report (IPCC , 2021). The link between these rapid modifications on the dynamics and frequency of weather patterns is an important question as it could have strong impact on society (e.g. Corti et al , 1999; Plaut and Simonnet , 2001; Pope et al , 2022).

The use of weather patterns to define similar atmospheric situations goes back to the early 1950s with for instance the
development of the Grosswetterlagen (Hess and Brezowsky , 1952), see also Barry and Perry (1973). Since then, such patterns



are mostly used to summarize the information content in weather forecasts (Neal et al , 2016), to develop statistical forecasting models (Nicolis et al , 1997; Vannitsem , 2001) or to investigate the quality of models (Davini and D'Andrea , 2020; Fabiano et al , 2020). The key advantage of such an approach is to reduce the complexity of the problem at hand by limiting the number of possible outcomes to a set of symbols that can be studied on their own.

The succession of weather patterns, as the underlying dynamics, displays a certain degree of randomness, mainly rooted in the natural property of sensitivity to initial conditions (e.g. Hannachi et al , 2017; Vannitsem , 2017). To describe such a dynamics a probabilistic approach is needed, which can be naturally cast in the context of the Information theory. This framework allows for the characterization of the predictability properties in terms of persistence, transition paths, and the degree of surprise of new patterns (Nicolis and Nicolis , 2012). Recently considerable progresses have been made in the

extension of the concept of information to dynamical systems out-of-equilibrium (Daems and Nicolis , 1999; Gaspard , 2004; Andrieux et al , 2007; Gomez-Marin et al , 2008; Roldán and Parrondo , 2010, 2012; Nicolis and Nicolis , 2012). Notably, the connections between information entropy, irreversibility and dissipation in such systems have been made, together with the impact of coarse-graining. Such developments open the way to analyze the dynamical and thermodynamical properties of non-equilibrium systems based on single coarse-grained trajectories.

The present work is devoted to investigating the dynamical properties on the succession of the North-Atlantic weather patterns as defined by Neal et al  (2016) in the observations and in the climate projections of the UK Met Office CMIP5 model (Pope et al , 2022), using recent tools of information theory. The focus is placed on the understanding of the impact of climate change on the predictability of the system through the evolution of the information entropy, and of the associated information entropy production. The work is organized as follows. The notions of information entropy and its production are

first introduced in Section 2. In Section 3, the data set used is briefly presented, and in Section 4, the results are discussed. Finally, a summary of the results are provided in the conclusions.

## 2   Information theory: Information entropy and entropy production

One key quantity introduced in the context of information theory is the (Shannon) entropy (Shannon , 1951),

$$S_I = -\sum_i p(i)\ln(p(i)) \tag{1}$$

where $p(i)$ is the probability to be in state $i$, with

$$\sum_i p(i) = 1. \tag{2}$$

This quantity (Eq 1) is an (weighted) average over the ensemble of states $i$ of a measure, $-\ln(p(i))$, of unexpectedness of an event (equivalent of the amount of information content in this event). This quantity has three important properties (Nicolis and Nicolis , 2012): (i) It is maximized when all the possible events have the same probabilities, like for instance in drawing

random numbers from a dice; (ii) adding an impossible event does not change $S_I$; and (iii) the additivity property, i.e. the entropy of a composite system $S_I(A, B) = S_I(A) + S_I(B|A)$.



The Shannon entropy used in this form is however static, and does not provide insight on the dynamics of the process. Other tools should therefore be used. A natural extension of this concept can be made to series of symbols, called *words*, known as the *block entropy*:

$$S_n = - \sum_{i_1, i_2, ..., i_n} p(i_1, i_2, ..., i_n) \ln(p(i_1, i_2, ..., i_n)) \tag{3}$$

where $p(i_1, i_2, ..., i_n)$ is the joint probability of the sequence $i_1, i_2, ..., i_n$. Block entropies have already been used to characterize the succession of weather patterns over Switzerland in Nicolis et al (1997). They showed in particular that this evolution is not a first-order Markov process that could, otherwise, be reduced to the analysis of the two-state transition matrix between successive patterns (Gardiner , 1996).

In 2004, Gaspard introduced the additional notion of time-reversed information entropy per unit time Gaspard (2004),

$$S_n^R = - \sum_{i_1, i_2, ..., i_n} p(i_1, i_2, ..., i_n) \ln(p(i_n, i_{n-1}, ..., i_1)) \tag{4}$$

where now the path through the different patterns is reversed in time, and the average is still performed along the forward path. If this quantity is subtracted to $S_n$, one gets the Kullback-Leibner divergence between the forward and backward trajectories in the form of

$$d_n = S_n^R - S_n = \sum_{i_1, i_2, ..., i_n} p(i_1, i_2, ..., i_n) \ln \frac{p(i_1, i_2, ..., i_n)}{p(i_n, i_{n-1}, ..., i_1)} \tag{5}$$

which is positive definite. This quantity is meant to characterize the time-asymmetry of the trajectory, and hence the irreversibility of the underlying process (Gaspard , 2004).

For $n$ tending to infinity, this quantity converges to an asymptotic value $d_\infty$, which is equal to the rate of contraction in phase space provided the partition is generated by a Markov process with infinitesimally small cells and infinitesimally small time steps (Gaspard , 2004; Nicolis and Nicolis , 2012), already demonstrating a strong connection of the information content with the underlying dynamics. At microscopic level, the quantity, $d_\infty$, can also be related to the physical entropy production under some appropriate assumptions (Andrieux et al , 2007; Gomez-Marin et al , 2008; Roldán and Parrondo , 2010, 2012). In this work, $d_\infty$ will be referred as the *information entropy production*.

In general, it can be shown that (e.g. Roldán and Parrondo , 2012):

$$d_\infty \geq ... \geq d_3 \geq d_2 \geq d_1 = 0 \tag{6}$$

and when the process is generating a first-order Markov dynamics, $d_2 = d_\infty$, which is readily available when computing the 2-state joint probabilities. When it is not first order Markov, $d_\infty$ can be accessed by computing the sequence of $d_k$, and using empirical laws, estimate $d_\infty$ (Roldán and Parrondo , 2010). In the current work, the process of succession of weather patterns is not first-order Markov (as discussed in the Appendix), and one must evaluate the sequence of lower bounds. This needs considerable data, and one can only estimate a small number of these lower bounds, yet providing very important information on the information entropy production.



The process of coarse-graining has also an impact on the amplitude of $d_k$ as shown in Gomez-Marin et al (2008), and also Gaspard (2022, personal communication): When reducing the number of variables to characterize the system, the amplitude of $d_k$ decreases.

## 3   Data

At the Met Office, 30 weather patterns were defined and are used on a daily basis in the operational forecasting suite in order to draw the overall evolution of the weather over the East part of the North-Atlantic and the western part of Europe (Neal et al , 2016). The evolution of the weather on a daily basis is available starting from January 1st, 1850 until now. In the present work, the series used start on January 1st 1850, until December, 31 2019, featuring 62,091 daily weather situations.

As the application of the tools mentioned in section 2 can be effectively used provided the number of data is large and the number of different patterns is small, it is important to find an appropriate balance between the length of the series and the number of patterns. 30 regimes is very large and already 900 entries have to be estimated for the 2-state joint probabilities, and this becomes even worse when increasing the length of the words. With such a small number of daily events, it is therefore unrealistic to keep a large amount of weather patterns. To solve that problem one can further cluster the patterns, as done for instance in Neal et al (2016) to 8 states. This number is still large to evaluate joint probabilities with such a small amount of data. We therefore further reduce the number of clusters to 6 with a merging of similar patterns, 8 with 6 and 7 with 5, as in Allen (2021). This can be even further reduced to 3, with the two dominant patterns 1 and 2, representing the positive and negative phases of the North Atlantic Oscillation (NAO), and the third one regrouping all other possible patterns (Allen , 2021). Besides their use for forecasting purposes, these data were used for different research purposes, such as for investigating the persistence of weather patterns Richardson et al (2018).

Besides the observational weather patterns, the UK Met Office produces the time series of weather patterns for a set of 15 different perturbed-parameter climate model versions under two climate scenarios (Pope et al , 2022). The two climate scenarios are defined based on two Representative Carbon Pathways (RCP), namely RCP 2.6 and RCP 8.5.

All simulations are run from December 1st 1899 to November 30th 2099, at a resolution of N216. The model is first forced by the historical forcing until 2005, and then forced with the Representative Carbon Pathway scenarios. More information can be found in Pope et al (2022). The simulations are projected on the same weather patterns as of Neal et al (2016), leading to a set of temporal evolution for the 8, 6 and 3 weather partitions defined above. For the simulations, $15 \times 71{,}970$ days are available.

## 4   Results

### 4.1   Information Entropy analysis of the observed data

The changes in the statistical and dynamical properties of the weather patterns is investigated through the analysis of the probabilities, the block-entropies and the information entropy production as a function of time. It should be first assumed that





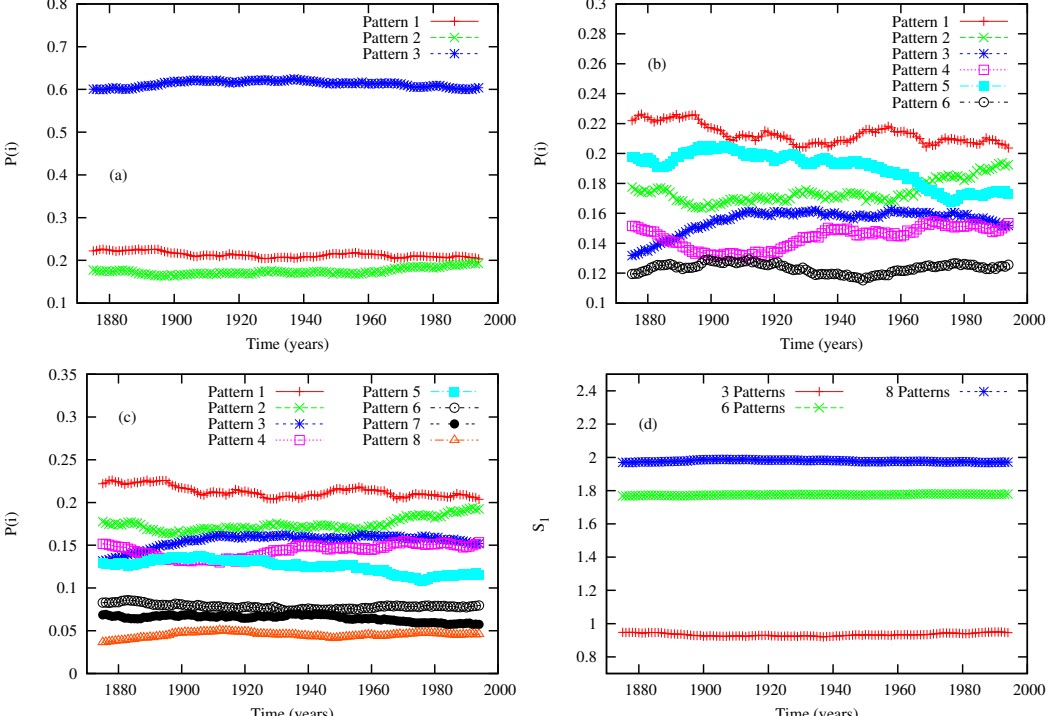

**Figure 1.** Temporal evolution of (a) the probability to be in one of the 3 patterns, (b) the probability to be in one of the 6 patterns, (c) the probability to be in one of the 8 patterns, and (d) the Shannon entropy for the three partitions.

the impact of climate change on the natural variability is slow, allowing to consider that the statistical properties of the weather are stationary during a sufficiently long period. Here a sliding window of 50 years has been defined for the evaluation of the

statistics, progressively moved forward in time. As the observation data are starting in 1850, the first period to consider is 1850-1899, then shifted forward in time every year. Note that the statistics will be associated (arbitrarily) to the 25th year of the window in the figures.

Figure 1 displays the probabilities for (a) 3, (b) 6, and (c) 8 partitions as a function of time. A clear trend in the probabilities is visible, and $\chi^2$ tests between the first and the last values indicate that the differences are highly significant. Panel (d) shows

the evolution of the Shannon entropy for the three partitions. Here however, the (static) information content does not change much as a function of time whatever the partition chosen.

The Shannon entropy, however, does not provide any information on the dynamics. Let us then turn to the dynamics of the weather patterns by investigating the 2-state entropy, $S_2$, providing information on the dynamics of the succession of pairs of patterns, together with the backward-in-time entropy, $S_2^R$, for the three partitions of interest (Figure 2). A first general remark

is the fact that $S_2$ is decreasing for most of the period, and then slightly increase, whatever the partition. This suggests that the information content decreases, with a less diverse set of pairs of events. When looking at the diagonal of the transition matrix,





$W(i|j)$, featuring the persistence from one day to the next, these conditional probabilities (for $i = j$) are increasing while $S_2$ is decreasing (not shown). This result suggests that the system becomes progressively more predictable and persistent during the historical period, except at the very end of the period.

On the same panels, $S_2^R$ is displayed, that would be larger than the $S_2$ if a time-asymmetry is present in the data (Gaspard , 2004; Andrieux et al , 2007). The amplitude of $S_2^R$ is indeed larger when considering the 6-pattern and 8-pattern partitions, but not for the 3-pattern partition. Considering the latter case first, this type of behavior suggests a time-symmetry (or detailed balance) of the dynamical process generated by that partition, as for instance found in the analysis of different alphabets used to "read" the DNA in Provata et al  (2014). Is this feature due to the fact that the series is too short or to a very specific feature

of this 3-pattern partition, remains to explore.

For the former cases of 6 and 8 patterns, the results are very interesting as the backward entropy is always much larger than the forward, suggesting a time-asymmetry related to the irreversibility of the process (Gaspard , 2004). This is further illustrated in panel (d) of Fig. (2), by the evolution of the difference between the backward and forward entropy, $d_2$. Note that $d_2$ shows an overall increase as a function of time. This would suggest an increase of the lower bound, $d_2$, of the information

entropy production over the North Atlantic. However, the trend is not reproduced when analyzing $d_3$ associated with the joint probabilities of 3 successive weather patterns, questioning the validity of the trend found with $d_2$. This type of analysis however suffers from a lack of data, that can only be compensated by investigating model runs. This point will be taken up further in the analysis of the UK Met Office models.

The analysis of the information entropies of pairs of events can be extended to longer blocks of symbols. In Fig. 3, a decrease

is also experienced whatever the length of the blocks of symbols (until 7 days), except at the end of the period. This feature is going in the same direction as for $S_2$, with an increase of predictability of the system. An additional indication of that is the constant decrease of the number of words (sequence of symbols) that are present in the window of 50 years as we move forward in time (not shown). In other words, the diversity of possible sequences is decreasing during the historical period. In panel (d), a different view of this evolution is displayed with the block entropy as a function of the length of the words at the

beginning of the historical period and at the end, further illustrating the change.

The analysis reveals a drastic modification of the dynamics of the succession of weather patterns over the North Atlantic and Western Europe, with an increase of predictability except at the end of the period. Is this feature a response to climate change or the presence of some low-frequency variability, is not clear at this stage. Another aspect that could affect the statistics is the number and quality of the observations used. A natural conjecture would be to believe that the first part of the period is not

much influenced by climate change, and therefore could reflect a natural tendency of the system provided that the dynamics is not affected much by the number and quality of observations. This conjecture could be challenged, either by analyzing weather pattern dynamics based on the same set of observation stations throughout the period, or with long reference runs of models allowing to clarify the impact of the low-frequency variability on the evolution of the dynamics of weather patterns.





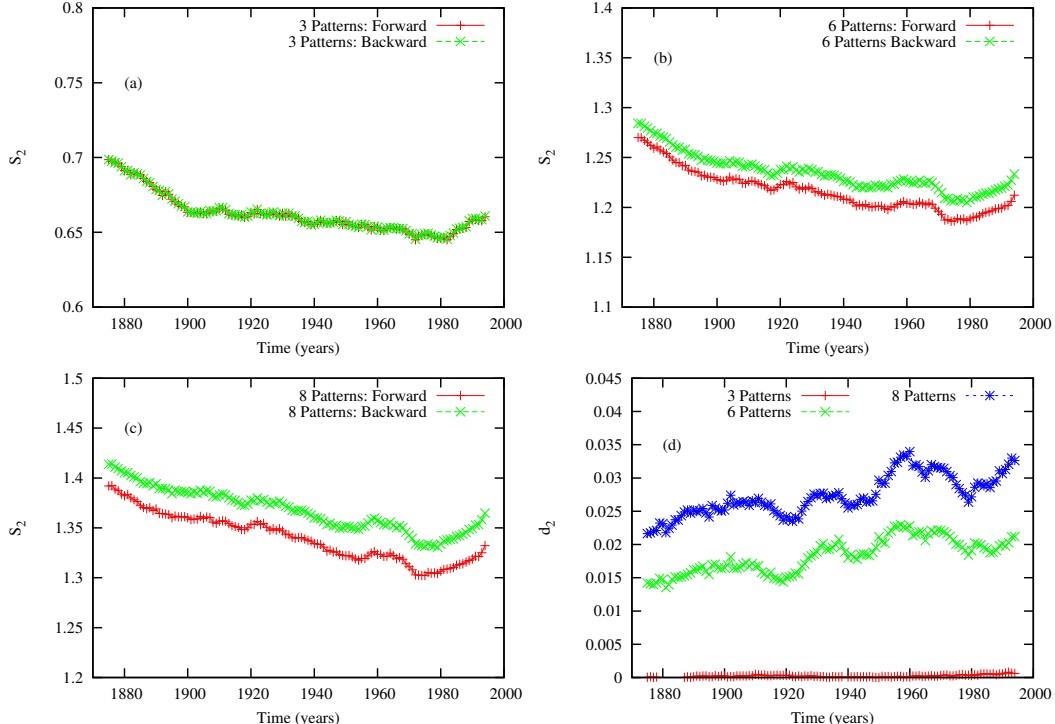

**Figure 2.** Temporal evolution of forward and backward entropies, $S_2$ and $S_2^R$, for (a) 3 patterns, (b) 6 patterns, and (c) 8 patterns. (d) the lower bound of the information entropy productio, $d_2$, for the three partitions.

## 4.2 Information Entropy analysis of the UK Met Office model

Figures 4 and 5 show the entropy, $S_2$, for the 15 model runs of the Met Office. The estimation based on the 15 realizations altogether is also shown (blue curve), together with the observations (red curve).

For the RCP2.6, $S_2$ shows a strong variability among the different realizations. One first remark is that $S_2$ for the different partitions over the overlapping periods is generally larger for the model runs than for the observations. This suggests that less regularities are present in the dynamics of the model with less predictability. Moreover, it is not clear at this stage whether the

large variability among the realizations is the impact of slightly different parameterizations within the model, or of a different possible realization of the dynamics starting from different initial states, or both.

For the RCP8.5, a similar picture is found, except that $S_2$ for most of the model runs shows larger values at the end of the 21st century.

In panel (d) of Figs. 4 and 5, the evolution of the lower bound, $d_2$, of the information entropy production of the model,

combined over the 15 different model versions, and of the observations, are displayed for the three partitions. The interesting message here is that $d_2$ is larger in the model than in the observations. This feature is also present for $d_3$ (not shown), which could reflect a larger information entropy production and degree of irreversibility than in reality.



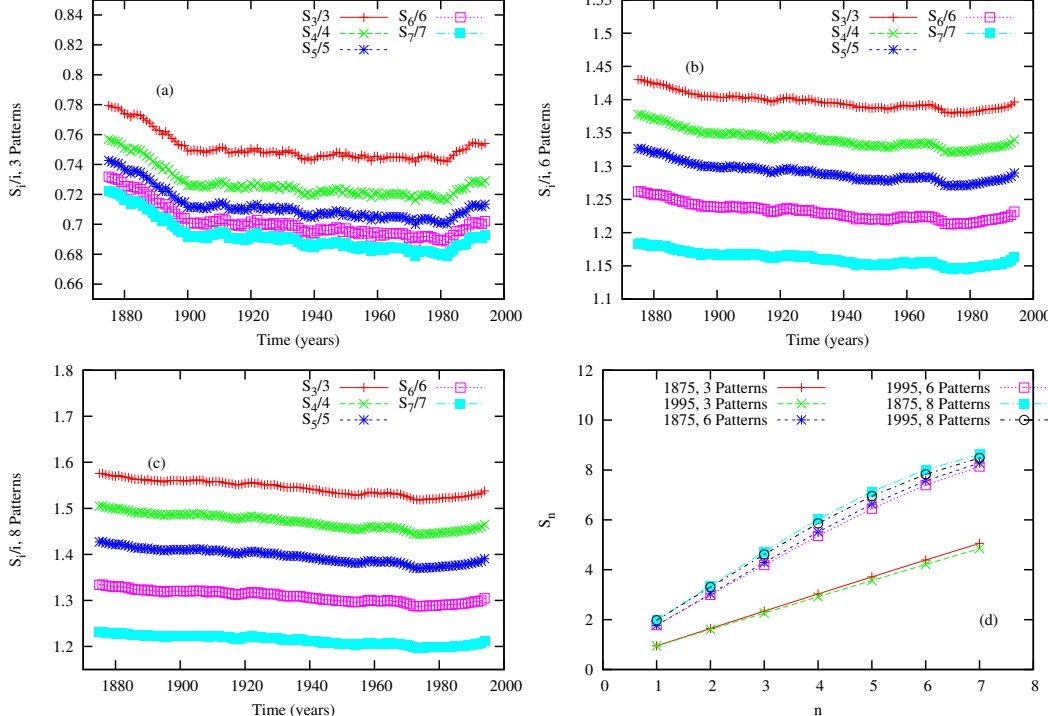

**Figure 3.** Temporal evolution of normalized forward block entropies, $S_i/i$, for (a) 3 regimes, (b) 6 regimes, and (c) 8 regimes. (d) Dependencies of the block entropies as a function of the length of the words, i, for the three partitions for the initial and final windows of 50 years of the observation dataset.

Finally in Figure 6, $d_2$, $d_3$ and $d_4$ are shown for both RCP scenarios. The larger bounds are not displayed as they are showing values smaller than the first three, in violation with their ordering, further indicating that the number of realizations are still not
sufficient to provide reliable estimates of high order joint probabilities. Interestingly, these three lower bounds are increasing as a function of time in both scenarios, suggesting that the information entropy production is also increasing as a function of time, and hence the atmospheric irreversibility. Furthermore, there is a change of the rate of increase of the lower bounds for one scenario or the other. In the case of the RCP8.5, the increase is faster around 2030 (corresponding to the period 2010-2050) indicating an acceleration of the change of information entropy production.

These remarkable results suggest that the information entropy production in the North Atlantic will considerably change depending on the type of scenario the Earth Climate system will follow. As the information entropy production is related to irreversibility of the dynamics, one may conjecture that production of heat and dissipation of the underlying dynamics will also increase substantially for large increase of the green-house gases. This is corroborated by works that have been done on the increase of dissipation on the North Atlantic region (Coumou and Rahmstorf , 2012) and (physical) entropy production in
general in climate models under climate change (Lucarini et al , 2010, 2011; Lembo et al , 2019; Kanno and Iwasaki , 2022).



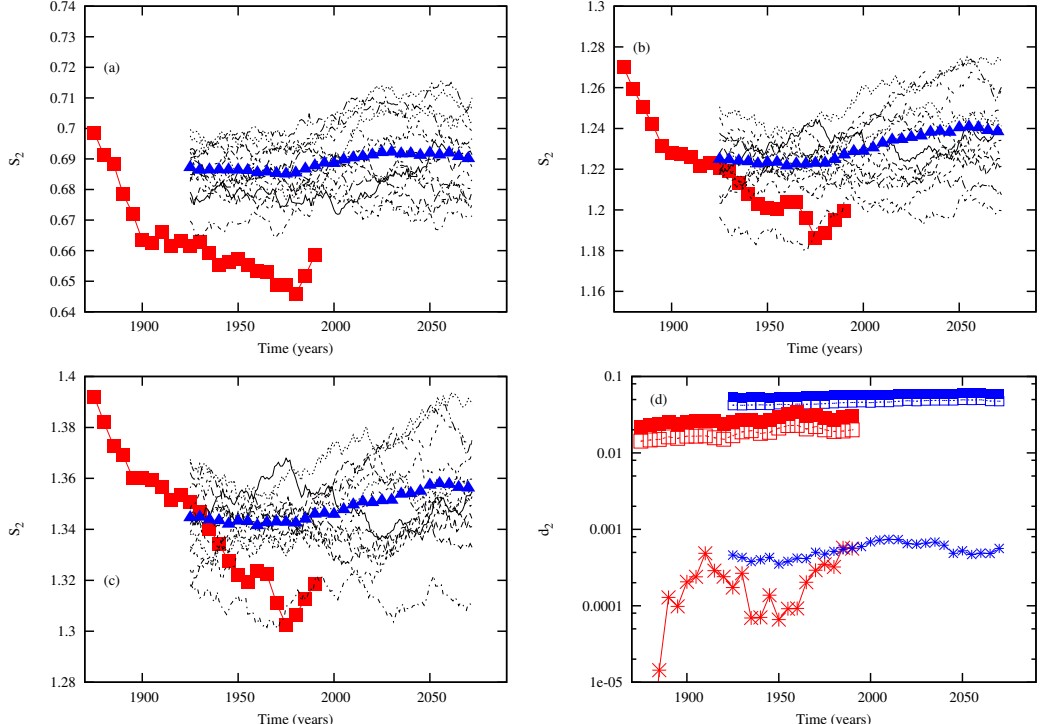

**Figure 4.** Temporal evolution of $S_2$ for the 15 model runs of the Met Office model from 1900 to 2099 under RCP2.6 scenario (black lines), for (a) 3 patterns, (b) 6 patterns, and (c) 8 patterns. The red curves represent the reference historical data and the blue triangle curve the average entropy over the 15 runs. (d) The lower bound of the information entropy production, $d_2$, for the three regime partitions for the observations (red), and for the average information entropy production of the model runs (blue). The different symbols (star, open square, full square) correspond to the three partitions 3, 6 and 8 patterns, respectively.

## 5 Conclusions

The dynamics of weather patterns over the North Atlantic under climate change is explored from the perspective of information theory with a focus on the information entropy and its production. The weather patterns are the ones defined by the Met Office (Neal et al , 2016), on which both the observations starting in 1850 and the model projections from 1900 to 2099, are projected.

Three sets of weather pattern partitions are used, 3, 6 and 8.

The first key message conveyed by this analysis is the overall decrease of the information entropy in the observations, except at the end of the period. This decrease indicates that the predictability increased during the historical period, with a slight decrease at the end. One key question is now to know whether this evolution is directly related to climate change, or to a natural low-frequency variability, or even to the change of the observational system over the North Atlantic. This question will

be addressed in the future.





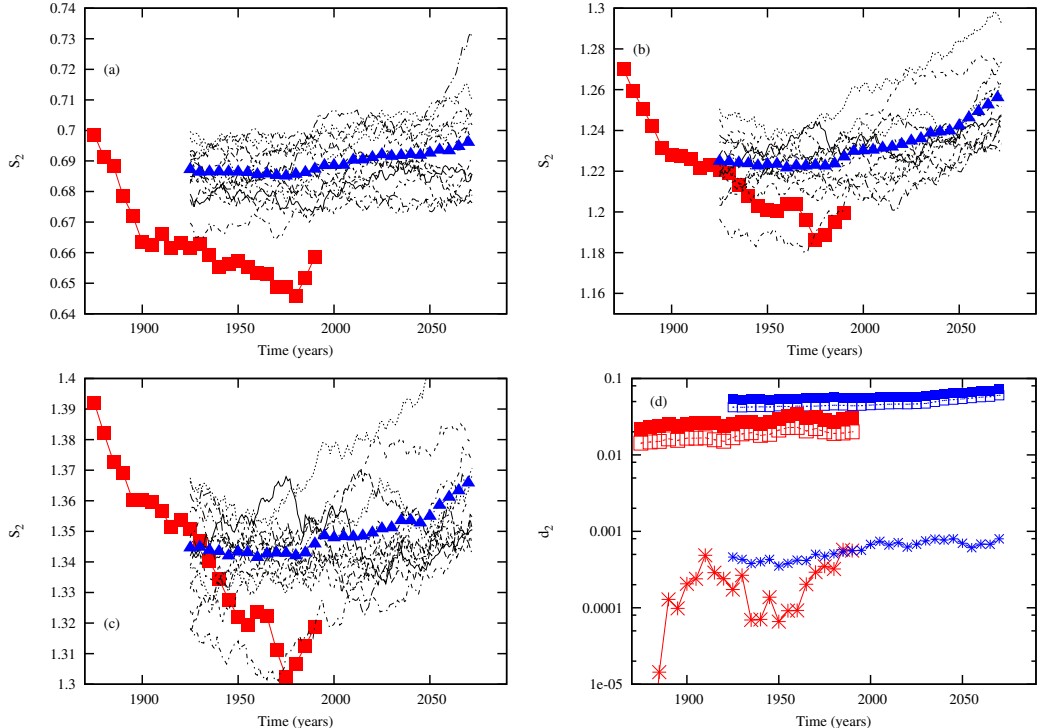

**Figure 5.** Temporal evolution of $S_2$ for the 15 model runs of the Met Office model from 1900 to 2099 under RCP8.5 scenario (black lines), for (a) 3 patterns, (b) 6 patterns, and (c) 8 patterns. The red curves represent the reference historical data and the blue triangle curve the average entropy over the 15 runs. (d) The lower bound of the information entropy production, $d_2$, for the three regime partitions for the observations (red), and for the average information entropy production of the model runs (blue). The different symbols (star, open square, full square) correspond to the three partitions 3, 6 and 8 patterns, respectively.

To further clarify the role of climate change in the evolution of the information entropy, the UK Met Office climate model runs under two climate scenarios, RCP2.6 and RCP8.5, were explored. The analysis of the Met Office climate model indicates that the information entropy is for most of the realizations larger than the one of the observations, suggesting a lower pre-dictability in the model. At the same time, all the realizations suggest that the information entropy will be larger by the end of

the 21st Century, further suggesting a decrease of weather pattern predictability.

The lower bounds of the information entropy production have been computed for both the observations and the model runs. For the observations, an increase of $d_2$ is found during the historical period, but the limited number of data does not allow to confirm this with the other bounds. For the model runs, these bounds are increasing as a function of time with a rate that depends on the specific scenario chosen, indicating an increase of the degree of irreversibility. This further allows us for making

the conjecture that heat production and dissipation associated with the emergence of irreversibility is increasing with climate change, corroborating earlier findings (Lucarini et al , 2010; Coumou and Rahmstorf , 2012; Lembo et al , 2019; Kanno and




**Figure 6.** Temporal evolution of the lower bounds of the entropy production $d_2$, $d_3$ and $d_4$ combining the statistics of the 15 Met Office model versions, under (a) RCP2.6 and (b) RCP8.5.



Iwasaki , 2022). As the rate of increase is much larger in the RCP8.5 scenario than in the RCP2.6 one, a further increase of heat production and dissipation should be expected under RCP8.5.

The novel approach of evaluating the (physical) entropy production based on coarse-grained time series at the microscopic level proposed by Gomez-Marin et al (2008); Roldán and Parrondo (2010, 2012) offers an important opportunity to estimate experimentally this quantity. Yet, when dealing with the dynamics of a macroscopic system like the atmosphere, the connection between the information entropy production and the physical entropy production is still missing. The possibility offered by these advances however opens the way to improve our knowledge of the dynamics of the climate system, provided appropriate researches are done in that direction.

The current model analysis is based on a set of slightly different model versions of the UK Met Office model. These differences could bias the estimates. A natural extension will be to explore large ensembles of a single model, and to explore different models of CMIP-class.

*Data availability.* The observation data set the historical classifications are available on request to the Met Office. Part of it can also be found on the Pangaea website at https://doi.pangaea.de/10.1594/PANGAEA.942896 (Neal, 2022). The UK Met Office Global model data used in
this study is all available from the Centre for Environmental Data Analysis http://data.ceda.ac.uk/badc/ukcp18/data.

## Appendix A: Markovianity of the succession of weather patterns

The Markovian nature of the dynamics can provide considerable simplifications in the description of coarse-grained dynamics. It is however now well known that lumping continuous state-space variables into a set of discrete states does not lead in general to a Markov dynamics (Nicolis and Nicolis , 2012). To check the Markovian character of the dynamics, statistical tests can be
performed. A test of the order of Markovianity has been proposed by Bilingsley (1961) and used in Provata et al (2014). This is a $\chi^2$ test under the hypothesis that the Markov chain is of order r:

$$\chi^2 = \sum_{i_1,...,i_s} \frac{[Np(i_1,...,i_s) - Np(i_1,...,i_{s-1})W(i_s|i_{s-r},...,i_{s-1})]^2}{Np(i_1,...,i_{s-1})W(i_s|i_{s-r},...,i_{s-1})} \tag{A1}$$

with a number of degrees of freedom of

$$N_F = N^s - N^{s-1} - (N^r - N^{r-1}) \tag{A2}$$

where $N$ is the number of successive times in the series, $W(i_s|i_{s-r},...,i_{s-1})$, the transition matrix from the path $i_{s-r},...,i_{s-1}$ to the new symbol $i_s$. The null hypothesis of the test is to assume that the process is Markov of order r. If the statistics of the test are larger than a certain threshold fixed by the level of confidence, then the null hypothesis is rejected. The test is applied to the data at our disposal with a level of confidence of 5%.

The test has been used for the 3, 6 and 8 patterns defined in Section 3. The results are shown in Table 1 for the observations.
This table indicates that whatever the number of patterns used here, they cannot be represented as first order Markov processes. It is however interesting to remark that when the number of patterns increases, the order of the Markov process needed





**Table A1.** Test $\chi^2$ for 3, 6 and 8 weather patterns for the series of observations. The two first columns represent the two Markov orders that are compared. The third column contains the number of degrees of Freedom of the test. The fifth and sixth columns contain the p-value of the test at the 5% level and the actual value of the test. If the actual value is smaller thant the p-value, the order $r$ is considered as the order of the Markov chain necessary to describe the dynamics of the weather patterns.

| | Order $r$ | $r+1$ | Numb Degrees Freedom | 5% p-value | Value of the test |
|---|---|---|---|---|---|
| | 0 | 1 | 4 | 9.5 | 39,432.5 |
| | 1 | 2 | 12 | 21.0 | 581.7 |
| 3 regimes | 2 | 3 | 36 | 51.0 | 272.3 |
| | 3 | 4 | 108 | 133.3 | 135.2 |
| | 4 | 5 | 324 | 367.0 | 281.6 |
| | Order $r$ | $r+1$ | Numb Degrees Freedom | 5% p-value | Value of the test |
| | 0 | 1 | 25 | 37.7 | 85,806.6 |
| | 1 | 2 | 150 | 179.6 | 1664.9 |
| 6 regimes | 2 | 3 | 900 | 970.9 | 1600.6 |
| | 3 | 4 | 5400 | 5572.1 | 3312.0 |
| | 4 | 5 | 32400 | 32819.8 | 9743.8 |
| | Order $r$ | $r+1$ | Numb Degrees Freedom | 5% p-value | Value of the test |
| | 0 | 1 | 49 | 66.3 | 108,108 |
| | 1 | 2 | 392 | 439.2 | 2,402.9 |
| 8 regimes | 2 | 3 | 3136 | 3267.4 | 3190.2 |
| | 3 | 4 | 25088 | 25457.6 | 6944.2 |
| | 4 | 5 | 200704 | 201747.0 | 18371.7 |

to represent properly the dynamics seems to decrease. This particular feature has however to be taken with caution as the computation of the probabilities of large blocks of symbols and with a large number of patterns, needs a large number of data much larger than the one currently at our disposal. A similar analysis has been performed for the different model runs with

similar conclusions.

*Author contributions.* The author did the analysis and wrote the manuscript.

*Competing interests.* The author declares no competing interest.



*Acknowledgements.* This work considerably benefited from discussions with C. Nicolis and P. Gaspard. P. Gaspard drew the attention of the author to the references of Roldán and Parrondo (2010) and Roldán and Parrondo (2012), which helped a lot in shaping the current findings.
We also thank R. Neal and J. Pope for providing the data, together with detailed information on their construction.





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
