# Peer review of "Weather pattern dynamics over Western Europe under climate change: Predictability, Information Entropy and Production"

_EGUsphere, 2022_

## Referee Comment (RC1)

Review on: Weather pattern dynamics over Western Europe under climate change: Predictability, Information Entropy and Production

Author: Stephane Vannitsem

Submitted to Nonlinear Processes in Geophysics, manuscript egusphere-2022-778

**General**

This study considers the atmosphere as a nonequilibrium steady state system and applies methods published by Gaspard (2004) to determine predictability and irreversibility by the information entropy in observational and simulated data. The author uses block entropies for the forward and the time reversed block entropy in time series describing the North Atlantic/West European weather. The time evolution is described as a coarse-grained sequence of visited boxes. The predictability is assessed by the forward entropy and the entropy production by the irreversibility due to time reversal asymmetry.

The data are Großwetterlagen in the Eastern North Atlantic/Western Europe sector, which had been extracted in observations and scenario simulations. The daily time series are reduced by clustering of the patterns to 3, 6 and 8 time series. The observational time period is 1850-2019, and the simulated data is for 1900-2100. As the numerical effort for the joint probabilities is enormous, the 30 patterns had to be drastically reduced to 3, 6 and 8. Furthermore, the block lengths had to be reduced to two, to calculate the entropy $S_2$. Thus, the present study is at the border of computational feasibility.

The study is insightful and relevant, although somehow preliminary, mostly due to computation restrictions. The agreement with previous studies hints at a reproducible core of results. The author should try to respond to the concerns, and if possible, less costly analyses might be added.

**Specific Comments**

I have several concerns, mostly on the use of Großwetterlagen and the nonstationarities in the data (mentioned in line 199).

1) **Großwetterlagen:** Großwetterlagen are certainly useful and have their merits in synoptics, but do they form a complete basis in state space? Do they depend on the domain similar to EOFs? Is a comparison with other sets meaningful? A short list of the selected Großwetterlagen patterns and the clusters would be useful.

2) **$S_2$ in the 3-pattern (Fig. 2):** For the 3-pattern the forward and backward entropies are $S_2^R=S_2$, hence there is no information entropy production in this basis. What does that mean for the choice of patterns? Is it possible to determine the entropy production independent of the basis?

3) **Anthropogenic climate change since 1860:** Global warming started early. Is it possible to find a similar behavior in the 21nd century, hence a common imprint of global warming?

4) **Natural low-frequency variability:** Is the sea surface temperature relevant for the frequencies of the patterns in Fig. 2?

5) **Decrease of $S_2$ during 1850-1900:** Is the strong decrease of $S_2$ in Fig. 2 a hint for an overlooked nonstationarity?

**Minor/Typos**

Line 10: is?

Lines 149-154: the paragraph could be clearer, is n=7?

Figure 3 caption: length of words, n?

Figures 4,5: a legend would be useful.

---

## Author Comment (AC1)

**Response to the points raised by Reviewer 1**

*Thank you very much for your evaluation of the manuscript. Please find below the list of your points together with the answers (italicized).*

**General**

This study considers the atmosphere as a nonequilibrium steady state system and applies methods published by Gaspard (2004) to determine predictability and irreversibility by the information entropy in observational and simulated data. The author uses block entropies for the forward and the time reversed block entropy in time series describing the North Atlantic/West European weather. The time evolution is described as a coarse-grained sequence of visited boxes. The predictability is assessed by the forward entropy and the entropy production by the irreversibility due to time reversal asymmetry.

The data are Großwetterlagen in the Eastern North Atlantic/Western Europe sector, which had been extracted in observations and scenario simulations. The daily time series are reduced by clustering of the patterns to 3, 6 and 8 time series. The observational time period is 1850-2019, and the simulated data is for 1900-2100. As the numerical effort for the joint probabilities is enormous, the 30 patterns had to be drastically reduced to 3, 6 and 8. Furthermore, the block lengths had to be reduced to two, to calculate the entropy $S_2$. Thus, the present study is at the border of computational feasibility.

The study is insightful and relevant, although somehow preliminary, mostly due to computation restrictions. The agreement with previous studies hints at a reproducible core of results. The author should try to respond to the concerns, and if possible, less costly analyses might be added.

*Thank you very much for the positive comments on the manuscript. Indeed, the approach proposed here has a very strong potential in the comparison between the properties of different models, providing clues on the properties of succession and occurrence of weather patterns. You will find below the answers to your specific points. The extension of the current analysis to other models and weather pattern projections are indeed worth performing. Here we focus on the weather patterns that were developed by the MetOffice, hence allowing for the direct use of their projections. An extension to other models or weather pattern projections would need to develop the full chain that was done at the MetOffice. This would request considerable efforts, that to my opinion, are beyond the scope of the present work. The current work should therefore be pursued in the future on theoretical and practical aspects. On the theoretical side, it is crucial to be able to relate the properties of the information entropy production to its thermodynamic counterpart at the macroscopic level, and on a practical side, to extend the analysis to very long model runs (forced or not) and to various decompositions in weather patterns.*

**Specific Comments**

I have several concerns, mostly on the use of Großwetterlagen and the nonstationarities in the data (mentioned in line 199).

- **Großwetterlagen:** Großwetterlagen are certainly useful and have their merits in synoptics, but do they form a complete basis in state space? Do they depend on the domain similar to EOFs? Is a comparison with other sets meaningful? A short list of the selected Großwetterlagen patterns and the clusters would be useful.

*The weather patterns are defined based on pressure fields over a very specific region covering western Europe and the eastern part of the Atlantic. So the description is limited to this specific region for large scale patterns. This is obviously limited but these patterns are very useful in developing scenarios for the weather evolution over this limited region. Some display strong similarities with patterns already isolated in other studies. See Hannachi et al (2017) for a review of former investigations.*

*The 8 weather patterns are described in Table 1 of Neal et al (2016), and displayed in their figure 3. The first two patterns with the largest populations in their analysis are referred to as the NAO+ (21.2 %) and NAO- (17.8 %) with opposite positive and negative mean sea level pressure anomalies over Iceland. These two patterns are usually found in the investigation of weather patterns over the North Atlantic and its surroundings. The other patterns mostly related to the local weather fields over the British Islands and western Europe are defined in Neal et al (2016) as Northwesterly, Southwesterly, Scandinavian high, High pressure centered over UK, Low close to UK and Azores high, respectively.*

*We have introduced this description in Section 3 where the description of the data is done:*

*"The 8 weather patterns are described in Table 1 of Neal et al (2016), and displayed in their figure 3. The first two patterns with the largest populations in their analysis are referred to as the NAO+ (21.2 %) and NAO- (17.8 %) with opposite positive and negative mean sea level pressure anomalies over Iceland. These two patterns are usually found in the investigation of weather patterns over the North Atlantic and its surroundings. The other patterns mostly related to the local weather fields over the British Islands and western Europe are defined in Neal et al (2016) as Northwesterly, Southwesterly, Scandinavian high, High pressure centered over UK, Low close to UK and Azores high, respectively."*

- **$S_2$ in the 3-pattern (Fig. 2):** For the 3-pattern the forward and backward entropies are $S_2^R = S_2$, hence there is no information entropy production in this basis. What does that mean for the choice of patterns? Is it possible to determine the entropy production independent of the basis?

*These two questions are indeed very interesting.*

*Concerning the first one, these values are close to equality suggesting a form of detailed balance (local equilibrium), for which the forward and backward time series structure seems statistically similar. This has also been found in Provata et al (2007) while analyzing very crude decomposition of symbols in the symbolic description of the human DNA. But one key aspect of coarse graining as done in the present work is the fact that the information entropy production is decreasing when the number of patterns decreases (see Figure 2d of the manuscript and Gomez-Marin et al (2008) for a theoretical explanation). This suggests that the almost equal values of $S_2$ and $S_2^R$ is most probably related to the statistical significance of the computed quantities. In order to get a better estimate of these quantities and to properly infer the information entropy production for such a coarse graining,*

*much longer time series is needed (or a set of realizations as obtained with the model runs). In order to better clarify this point, we have modified the last paragraph of Section 2 as:*

*"The process of coarse-graining has also an impact on the amplitude of d_k as shown in Gomez-Marin et al (2008), and also Gaspard (2022, personal communication): When reducing the number of symbols or patterns to characterize the evolution of the system, the amplitude of d_k decreases. This could lead to estimates for very coarse partitions of the dynamics that are not statistically very well defined for the information entropy production. This is most probably the case of the analyses that are done with the coarser partition for the observations below. This problem is however alleviated when investigating the set of model runs as better statistics can be obtained."*

*For the second question, the information entropy production based on coarse-graining is indeed dependent on the partition made. This is clearly visible by increasing the number of clusters in our analysis. The only general statement that can be made is obtained when the number of clusters is tending to infinity. That would imply that the cells of the partition are infinitesimally small, hence converging toward the continuous set of equations. In this situation and for a large class of dissipative dynamical systems obeying a Fokker-Planck equation, the information entropy production converges toward minus the sum of the Lyapunov exponents (e.g. Nicolis and Nicolis, 2012). In the context of the atmospheric dynamics and the very short series available, this limit is far from reachable.*

- **Anthropogenic climate change since 1860:** Global warming started early. Is it possible to find a similar behavior in the 21nd century, hence a common imprint of global warming?

*This is indeed a very good point, that could also be related to the question of the impact of low-frequency variability on the information entropy. Investigating what was happening before 1850 would be indeed very useful. As said earlier in this response, this can be achieved but the whole process of projection should be made using new model trajectories, in particular long runs on millennial time scales. This will not be addressed here, as we do not have access to such data yet.*

- **Natural low-frequency variability:** Is the sea surface temperature relevant for the frequencies of the patterns in Fig. 2?

*This is also a possible conjecture that should be clarified in the future by investigating long runs with fixed anthropogenic forcing, in such a way to isolate the internal dynamics of the model from the external forcing. We cannot address this aspect at this stage of the research.*

- **Decrease of $S_2$ during 1850-1900:** Is the strong decrease of $S_2$ in Fig. 2 a hint for an overlooked nonstationarity?

*This question is an important one that was also mentioned in our conclusions. Is this decrease related to the intrinsic low-frequency variability, or to natural fluctuations of the statistics, or to an increase of the quality of the measurement network? This remains to be discussed by analyzing, for instance, long model runs on millennial time scales.*

**Minor/Typos**

Line 10: is?

*Thank you very much.*

Lines 149-154: the paragraph could be clearer, is n=7?

*We have tried to make it more clear. Indeed i=7 days.*

Figure 3 caption:  length of words, n?

*Thank you very much for pointing this out. We have changed the 'n' in panel (d) into an 'i' in order to be consistent with the other panels.*

Figures 4,5: a legend would be useful.

*We keep the panels without legends in order to have less crowded panels. The details are provided in the caption.*

**References**

Gaspard, P., 2004: Time-reversed Dynamical Entropy and Irreversibility. J. Stat. Phys., 117, 599-615.

Gomez-Marin, A., J. M. R. Parrondo, and C. Van den Broeck, 2008: Lower bounds on dissipation upon coarse graining. Phys. Rev. E 78, 011107.

Hannachi, A., Straus, D. M., Franzke, C. L. E., Corti, S., and Woollings, T. (2017), Low-frequency nonlinearity and regime behavior in the Northern Hemisphere extratropical atmosphere, Rev. Geophys., 55 , 199– 234.

Neal, R., D. Fereday, R. Crocket and R. Comer, 2016: A flexible approach to defining weather patterns and their application in weather forecasting over Europe. Meteorol. Apps., 23, 389-400.

Nicolis, G. and C. Nicolis, 2012: *Foundations of Complex Systems*. World Scientific, Singapore.

Provata, A., C. Nicolis and G. Nicolis, 2014: DNA viewed as an out-of-equilibrium structure. Phys. Rev. E, 89, 052105.